# Juvenile Idiopathic Arthritis-Associated Chronic Uveitis: Recent Therapeutic Approaches

**DOI:** 10.3390/jcm10132934

**Published:** 2021-06-30

**Authors:** Pierre Quartier

**Affiliations:** 1Pediatric Immunology, Hematology and Rheumatology Unit, Necker-Enfants Malades Hospital, Assistance Publique-Hopitaux de Paris, 75015 Paris, France; pierre.quartier@aphp.fr; 2RAISE Reference Centre for Rare Diseases, IMAGINE Institute, Necker-Enfants Malades Hospital, Assistance Publique-Hopitaux de Paris, 75015 Paris, France; 3INSERM Unit 1163, Université de Paris, 75005 Paris, France

**Keywords:** uveitis, juvenile idiopathic arthritis, methotrexate, biologic treatment, adalimumab, abatacept, tocilizumab

## Abstract

Pediatric patients with early onset (before the age of 6 years), antinuclear antibody positive, oligoarticular or polyarticular juvenile idiopathic arthritis (JIA), and some children with no arthritis may develop chronic, anterior uveitis. Recent recommendations insist on the need to perform slit lamp examination every 3 months for at least 5 years in early onset JIA patients in order to diagnose uveitis before complications develop. Local steroid therapy is usually the first-line treatment. However, in patients requiring steroid eye drops for several months, systemic immunomodulatory therapy is indicated. Methotrexate (MTX) is then prescribed in most cases; however, some patients also need anti-tumor necrosis factor alpha monoclonal antibody therapy and, in some cases, other biologics to control uveitis and avoid long-term ocular damage. Expert ophthalmologists and pediatricians must be involved in taking care of such patients. Immunomodulatory treatment must not be too easily interrupted and may even be intensified in some cases, particularly if there is a need for optimal disease control before ophthalmologic surgery. In good responders to MTX and/or biologics, treatment must be maintained at least 1 year, possibly even 2 years after achieving remission before tapering treatment intensity.

## 1. Introduction

Juvenile idiopathic arthritis (JIA) is a disease characterized by arthritis starting before the age of 16 years, lasting at least 6 weeks, and with no cause identified [1]. Two-thirds of JIA patients present early onset JIA, starting before the age of 6 years, with either a polyarticular or more often an oligoarticular (i.e., less than five arthritis within the first 6 months) joint involvement pattern and, in most cases, presence of non-specific antinuclear antibody (ANA) [2]. These ANA-positive, early onset JIA patients carry a risk of 20–30% chance of chronic, anterior uveitis, which almost always starts within the first 5 years of the disease. This uveitis is initially asymptomatic, without eye redness, and affects both eyes in 70–80% of these patients within 1 year. Some children with no arthritis but, in most cases, ANA, may develop the same chronic, anterior uveitis.

While the development of new treatments has markedly improved the prognosis of arthritis in JIA patients, chronic uveitis may still, in the absence of early detection and effective treatment, lead to severe complications and visual impairment in a significant proportion of patients [3,4,5].

In patients with another category of JIA, enthesitis-related JIA/juvenile spondylarthritis, which usually starts after the age of 6 years and mainly affects male patients, recurrent acute anterior uveitis may develop, similarly as in adult-onset spondyloarthritis. As it is associated with marked clinical symptoms, including eye redness, there is usually no diagnosis delay and the visual prognosis is much better than in ANA positive patients who develop chronic anterior uveitis [5].

In this article, we will focus on chronic, anterior, “white eye” JIA-associated uveitis.

Several international and national recommendations guide the treatment of JIA-associated chronic anterior uveitis [6,7,8]. While indicating some key recommendations from the European SHARE initiative [6] and the most recent American College of Rheumatology (ACR) recommendations [7], we will mainly refer to the most recent French recommendations for pediatric and adult-onset, chronic, non-infectious uveitis [8].

## 2. Classical Therapeutic Approaches in JIA-Associated Chronic Uveitis

Local corticosteroid treatment is indicated in the first line to control anterior uveitis [6,7,8]. However, while steroid eye drops, in most cases dexamethasone or prednisone acetate 1%, are currently used, corticosteroid injections should be avoided if possible in children and young adults [8]. In addition to corticosteroid eye drops, other local treatments may be administered, such as mydriatics, to reduce the risk of synechiae or treatments to control ocular hypertonia.

In children and adolescents with JIA and controlled uveitis who are tapering or discontinuing topical glucocorticoids, ophthalmologic monitoring within 1 month after each change of topical glucocorticoids is strongly recommended over monitoring less frequently [7].

In patients whose uveitis requires several months of local steroid therapy, the risk of side effects has led several panels of experts to recommend the introduction or change of systemic therapy rather than maintaining long-term local steroid treatment [6,7,8]. According to the SHARE recommendations, systemic immunosuppression is recommended if inactivity could not be reached within 3 months or inflammation is reactivating during steroid dose reduction [6]. In France, according to the most recent recommendations from the “Protocole National de Diagnostic et de Soins” for chronic, non-infectious uveitis, it is recommended in both adults and children to consider the introduction of systemic immunosuppressive/immunomodulatory treatment if a patient discloses active anterior uveitis and is intolerant to local treatment or requires two drops of dexamethasone (or equivalent) per eye over 6 months or three drops over three months (Table 1) [8]. Systemic steroid administration is avoided unless there is an urgent need to very efficiently control active uveitis, particularly in the context of ocular surgery (Table 2).

In JIA patients with chronic anterior uveitis that is controlled by systemic therapy but remain on 1–2 drops/day of prednisolone acetate 1% (or equivalent), tapering topical glucocorticoids first is strongly recommended over systemic therapy [7].

In JIA patients with chronic, anterior uveitis, methotrexate is recommended as the first-line systemic immunomodulatory treatment, at a dosage between 10 and 15 mg/m^2^ or between 0.3 and 0.6 mg/kg once a week, maximum 25 mg/week, orally or subcutaneously. This treatment is well tolerated in most cases; however, some patients develop intolerance, sometimes hematologic toxicity, more often hepatic or digestive intolerance, with a significant proportion of children becoming more and more disgusted over months or years by the drug and gradually unable to cope with oral or even subcutaneous MTX administration.

In addition, MTX is a relatively slow-acting drug; it usually takes 3 months or so to achieve its full efficacy, and it is not always effective enough to control uveitis and reduce local steroid dependency.

## 3. Anti-Tumor Necrosis Factor (TNF) Alpha Antibody Treatment in JIA-Associated Chronic Uveitis

By contrast with the soluble Tumor Necrosis Factor (TNF) receptor etanercept, which is an effective treatment of arthritis but not uveitis, anti-TNF monoclonal antibodies are in most cases effective in patients with idiopathic or JIA-associated chronic, anterior uveitis.

In patients with an inadequate response to local steroid therapy and methotrexate (MTX), two placebo-controlled, double-blind randomized trials, the large multicenter SYCAMORE trial in the United Kingdom and the smaller ADJUVITE trial in France, demonstrated the efficacy of the anti-TNF antibody adalimumab [9,10]. A significant effect of adalimumab on uveitis was documented as early as 2 months in most patients when using sensitive assessment tools such as laser flare photometry [9]. It is recommended to use adalimumab at a dosage of 20 mg every 2 weeks in patients weighing less than 30 kg, 40 mg every 2 weeks in the others, in combination with MTX, unless the patient is intolerant to MTX.

For several experts, in JIA patients with severe active chronic anterior uveitis and sight-threating complications, starting methotrexate and a monoclonal anti-TNF antibody immediately is conditionally recommended over methotrexate as monotherapy [7,8]. However, some patients with JIA-associated uveitis do not respond to adalimumab or flare.

The development of anti-adalimumab neutralizing antibodies is one of the main causes of secondary treatment failure. To minimize the risk of anti-biomedication neutralizing antibodies, the association to MTX or another immunosuppressive drug may be useful; however, it is above all crucial not to use low-dose adalimumab or miss adalimumab injections. In this regard, it is inappropriate to interrupt adalimumab in case of surgery or mild infection [8]. In some cases, it might even be worth considering higher adalimumab doses or weekly injections (Table 2) [6,7,8].

Other anti-TNF monoclonal antibodies have been used as first biologic therapy or, more often, in patients who failed to respond or maintain adequate response to adalimumab. Although no placebo-controlled trials have been performed, intravenous infliximab, sub-cutaneous golimumab, and other anti-TNF alpha monoclonal antibodies are certainly effective [11,12]; however, anti-biomedication-neutralizing antibodies may develop for all of them. One of the ACR recommendations is that in JIA patients with active chronic anterior uveitis who have an inadequate response to one anti-TNF monoclonal antibody at standard JIA dose, escalating the dose and/or frequency to above-standard is conditionally recommended over switching to another anti-TNF monoclonal antibody [7]. However, this recommendation is only valid in the absence of anti-biomedication-neutralizing antibodies [6,7,8].

## 4. Other Therapeutic Options

In accordance with available evidence-based medicine, one of the ACR recommendations is that in JIA patients with active chronic anterior uveitis who have failed a first anti-TNF monoclonal antibody at an above-standard dose and/or frequency, changing to another anti-TNF monoclonal antibody is conditionally recommended over a biologic in another category [7]. According to the SHARE initiative, the use of anti-TNF treatment strategies (adalimumab > infliximab > golimumab) is recommended in patients with uveitis refractory/resistant to methotrexate, and switching between different anti-TNF treatments might be valuable if uveitis is refractory to the first anti-TNF [6].

However, several observations suggest that tocilizumab, an anti-interleukin (IL)-6 receptor monoclonal antibody which is effective in polyarticular or systemic JIA on both systemic symptoms and arthritis [13,14], is also effective in some patients with chronic, non-infectious uveitis, particularly on refractory uveitis-associated macular edema [15]. There are similar observations for the anti-interleukin (IL)-6 receptor monoclonal antibody sarilumab.

A multicenter, single-arm phase II trial, APTITUDE, recently assessed the safety and efficacy of tocilizumab in children with JIA-associated uveitis refractory to both MTX and adalimumab [16]. Tocilizumab was administered subcutaneously at a fixed dose of 162 mg every 4 weeks in children weighing less than 30 kg and every 2 weeks in the other patients.

The primary endpoint of the trial was not met as only 7 out of 21 patients achieved a significant improvement of anterior chamber inflammation according to the Standardization of Uveitis Nomenclature (SUN) criteria [17] after 12 weeks. In addition, only six patients could reduce the number of steroid eye drops to less than two per eye per day. However, the authors considered that tocilizumab was beneficial in a subset of patients; in particular, macular edema that was present in four patients at baseline resolved in three cases. This is certainly of interest as only difficult-to-treat patients were included in this trial, as their uveitis was refractory to MTX and adalimumab therapy. It is noteworthy that in this trial, tocilizumab was used at the same dose as in children with polyarticular-course JIA, and no dose increase was allowed. In such difficult-to-treat patients, using twice more frequent injections, as is recommended in patients with systemic-onset JIA, or allowing to increase the dose in non- or partial responders would have been worth considering [18].

Of note, in both SYCAMORE and APTITUDE trials, most patients had marked chronic inflammation at inclusion, as assessed by the presence of high numbers of cells in the anterior chamber on slit-lamp examination. Thanks to an increased awareness of JIA-associated uveitis and earlier treatment, many patients in our countries have no cells on slit-lamp examination; however, their uveitis may be active and deserve more effective therapy, as demonstrated in the ADJUVITE trial when assessing uveitis activity by laser flare photometry [10].

In addition to anti-TNF or –IL-6 receptor antibody treatments, other treatments have been or are being tested in patients with JIA-associated chronic anterior uveitis.

Cytotoxic T-lymphocyte-associated protein 4 immunoglobulin (CTLA-4Ig) treatment with abatacept, which has been approved in the treatment of JIA patients with a polyarticular course, also seems effective in some patients with chronic uveitis, although no controlled trial has been performed [19].

Data for rituximab and Il-1 inhibitors are even more limited. Nevertheless, one of the SHARE recommendations is that tocilizumab, rituximab, and abatacept might be potential options for cases refractory to previous anti-TNF therapy [6].

The oral Janus Kinase (JAK)-1/2 inhibitor baricitinib is being tested in patients with ANA positive, early onset, idiopathic or JIA-associated uveitis in an ongoing international trial (NCT04088409).

## 5. Some Peculiar Situations

In patients with early onset, non-systemic, ANA positive oligo or polyarticular JIA with active arthritis on MTX requiring biologic therapy, several pediatric rheumatologists may not feel at ease to introduce etanercept, which is a very effective treatment of arthritis but has no effect on uveitis, unless they are sure that the child will benefit from a proper follow-up by an ophthalmologist, that should involve slit-lamp examination every 3 months during the first 5 years of the disease, in order to detect at an early stage the possible occurrence of uveitis [8]. This is particularly true in the patients who have to stop MTX for intolerance, as MTX most likely has some preventive effect regarding the occurrence of uveitis. In such patients, if no proper, frequent ophthalmologic assessment is possible, an anti-TNF antibody treatment, such as adalimumab, or the CTLA-4Ig abatacept may be privileged, as these biologics are most likely to prevent, to a large extent, the risk of developing uveitis.

In patients with active chronic, anterior uveitis who do not respond well and/or develop complications to topical corticosteroid therapy, particularly in case of ocular hypertonia deserving quick, effective control of inflammation and minimal usage of corticosteroids, it might be worth discussing the early introduction of anti-TNF antibody treatment in association with MTX to control uveitis as quickly as possible, using an expert-center driven treat-to-target approach, an approach that is now encouraged in the treatment of JIA patients [20].

In patients who do not respond well to adalimumab, checking treatment compliancy, dosing adalimumab serum level and anti-drug antibodies are recommended before increasing the dosage or moving to another treatment (Table 2).

In JIA patients with both arthritis and uveitis whose uveitis has been well controlled on adalimumab (and MTX if tolerated) but arthritis remains active, dosing adalimumab serum level and considering weekly injections is an option, moving to another biologic such as infliximab or an anti-IL-6 receptor antagonist that might be more effective on arthritis while also active on uveitis could be another option.

In patients with chronic anterior uveitis and marked macular edema, anti-IL-6 receptor monoclonal antibody treatment is also worth considering.

In patients whose uveitis remains active and who need ocular surgery, it is important to minimize the risk of complications by intensifying uveitis therapy shortly before and after surgery. This might include the use of pulsed methylprednisolone perfusions and/or an increase of biologic treatment intensity, e.g., by transiently moving from every other week to weekly adalimumab injections.

In patients with long-lasting uveitis and arthritis remission on adalimumab and MTX but no local or systemic steroids, the treating physicians may consider or not tapering treatment intensity or withdrawing adalimumab after 1 year of uveitis remission off steroids while carefully following the patient [8]. Both the ACR and the European SHARE recommendations are in favor of at least 2 years of quiescence before tapering treatment intensity when uveitis is well controlled on Disease-Modifying Anti-Rheumatic Drug (DMARD) and systemic biologic therapy only [6,7]. Earlier treatment withdrawal or treatment intensity reduction is at high risk of uveitis flare. The ongoing randomized, controlled ADJUST trial (https://clinicaltrials.gov/ct2/show/NCT03816397 (accessed on 13 May 2021)) aims to assess the efficacy of discontinuing adalimumab after demonstrating control of JIA-associated uveitis for at least 12 months. There is clearly a need for biomarkers to predict the risk of flare following treatment discontinuation.

## 6. Conclusions

JIA-associated chronic uveitis needs to be detected as early as possible by encouraging trimestrial slit-lamp examination in the patients at risk. Although local treatment using corticosteroid eye drops is sufficient in a proportion of uveitis patients, wider use of methotrexate and biologics such as anti-TNF alpha monoclonal antibodies has been associated with marked improvement in controlling uveitis activity in more severe or persistently active cases. New tools such as laser flare photometry are worth considering to properly assess uveitis activity and follow treatment efficacy. In all these patients, it is crucial to have a constant dialogue between the ophthalmologist and the pediatric rheumatologist, the rheumatologist, or the internal medicine specialist to offer the best possible care and lower the burden of ocular complications and visual loss in adulthood.

## Figures and Tables

**Table 1 jcm-10-02934-t001:** Chronic, JIA-associated, non-infectious uveitis: recent key recommendations.

Situation	Recommendation *
(1) JIA patient at risk of chronic uveitis	Slit-lamp examination every 3 months for 5 years
(2) Recent diagnosis of chronic uveitis, a need of expertise	Early involvement of expert ophthalmologists and pediatricians
(3) Recent diagnosis of chronic uveitis systematic diagnosis screen	Contact experts to avoid unnecessary exams
(4) Chronic anterior uveitis and need of ≥2 drops of dexa drops over 6 months or ≥3 drops over 3 months (cumulated)	Avoid local steroid injections, rather, discuss systemic immunomodulatory treatment (such as MTX)
(5) Chronic anterior uveitis active after 3 months of MTX and local steroid therapy	Discuss anti-TNF antibody therapy (adalimumab authorized in this indication)
(6) Chronic uveitis in remission on adalimumab (±MTX) treatment	Do not stop adalimumab before at least 1 year of complete remission

JIA, Juvenile Idiopathic Arthritis; Dexa, dexamethasone; MTX, methotrexate; TNF, tumor necrosis factor. * French recommendations (https://www.has-sante.fr/jcms/p_3187246/fr/uveites-chroniques-non-infectieuses-de-l-enfant-et-de-l-adulte (accessed on 5 June 2020)).

**Table 2 jcm-10-02934-t002:** Chronic, non-infectious uveitis: some peculiar situations.

Situation	Recommendation *
(1) Uveitis remaining very active or flaring on well-conducted local treatment	Introduce MTX therapy ± short oral steroid course or introduce MTX + anti-TNF antibody treatment
(2) Marked macular edema on MTX and adalimumab therapy	Replace adalimumab with anti-IL-6 receptor monoclonal antibody treatment
(3) Cataract requiring surgery in a patient with active uveitis	Intensify pre- and post-surgery systemic therapy: ±IV corticosteroids ± anti-TNF antibody treatment
(4) Measle epidemy, non-immunized child on MTX and adalimumab treatment	Discuss anti-measle, life-attenuated vaccination without interrupting ongoing treatment *
(5) Partial response to adalimumab and low adalimumab concentration without neutralizing antibodies	Check compliancy to treatment, consider weekly adalimumab injections, repeat dosage and anti-adalimumab-eutralizing antibodies testing

MTX, methotrexate; TNF, tumor necrosis factor; IL, interleukin; IV, intravenous. * Although life-attenuated vaccines are theoretically contraindicated on MTX and/or biologic treatments, the risk of adverse reaction is almost zero in the absence of primary immunodeficiency. On the other hand, measle on MTX and adalimumab may be life-threatening, and interrupting MTX and adalimumab is at risk of uveitis flare. There should be a discussion between experts and the patient to consider the benefit/risk balance, which may be in favor of performing a life-attenuated vaccination without interrupting MTX and adalimumab treatment.

## Data Availability

Not applicable.

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
