# Peer review of "Juvenile Idiopathic Arthritis-Associated Chronic Uveitis: Recent Therapeutic Approaches"

_jcm, 2021, doi:10.3390/jcm10132934_

Round 1

Reviewer 1 Report

Table 1: please reformulate the paragraph "Recent diagnosis of chronic uveitissys-tematic diagnosis screen"

page 3 line 99: change to 30 Kg

pg 5 line 177: I suggest to change to "dosing adalimumab serum level and anti-drug antibodies"

pg 5 line 201: change bee to been

Author Response

Table 1: please reformulate the paragraph "Recent diagnosis of chronic uveitis sys-tematic diagnosis screen"

I agree it had to be reformulated as it appears in the revised version: Recent diagnosis of chronic uveitis, a need of expertise

Page 3 line 99: change to 30 Kg

I changed to 30 kg, thank you.

pg 5 line 177: I suggest to change to "dosing adalimumab serum level and anti-drug antibodies"

I changed accordingly.

pg 5 line 201: change bee to been

Change made in the revised version

Reviewer 2 Report

This review describes recent advances in the treatment of JIA-associated uveitis. It provides a brief but comprehensive overview of French treatment guidelines and recommendations that the author has also contributed to.

  1. As this journal and article is meant for a general international audience, the author solely references their French uveitis guidelines. Are the French guidelines also subject to peer-review and publication in a peer-reviewed journal? Additionally, the version referenced is a non-English version, which makes it difficult for the reviewer to interpret the reference. If authors solely wish to refer to the French guidelines, this should be stated in the title of the manuscript and/or abstract.
  2. In terms of steroid eyedrops, certainly dexamethasone may be used, but there are are topical steroids that should be mentioned as alternatives, especially since other countries, prednisone acetate 1% may be frequently used instead of dexamethasone.
  3. Why stop immunosuppression after at least 1 year of quiescence? Although a low level of evidence/conditional recommendations, several other guidelines refer to at least 2 years. Authors should also reference the ongoing ADJUST trial (https://clinicaltrials.gov/ct2/show/NCT03816397).
  4. For the situation of “Marked macular edema on MTX and adalimumab therapy,” authors recommend replacing adalimumab by anti-IL-6 treatment, why not try infliximab?
  5. In general, do authors consider the use of other anti-TNF agents (eg infliximab) this should be discussed as the current portrayal of the article suggests that switching to anti IL-6 is the next step after adalimumab/MTX failure.

Author Response

Manuscript jcm-1262784 Responses to Reviewer 2:

This review describes recent advances in the treatment of JIA-associated uveitis. It provides a brief but comprehensive overview of French treatment guidelines and recommendations that the author has also contributed to.

  1. As this journal and article is meant for a general international audience, the author solely references their French uveitis guidelines. Are the French guidelines also subject to peer-review and publication in a peer-reviewed journal? Additionally, the version referenced is a non-English version, which makes it difficult for the reviewer to interpret the reference. If authors solely wish to refer to the French guidelines, this should be stated in the title of the manuscript and/or abstract.

I thank the reviewer for his comments and agree the review was to focused on the recent French guidelines, which have not yet been published in a peer-reviewed journal.

I tried to better acknowledge the fact that key recommendations have been proposed by several panels of experts in different part of the worlds before focusing on French recommendations. Therefore:

The abstract was modified to better reflect recent recommendations from the European SHARE initiative (ref 6) and the American College of Rheumatology (ref 7)

At the end of the introduction, page 2, I put a paragraph that replaces a less complete statement that used to be in section 2 and did not state as clearly the following: “Several international and national recommendations guide the treatment of JIA-associated chronic anterior uveitis [6-8]. While indicating some key recommendations from the European SHARE initiative [6] and the most recent American College of Rheumatology (ACR) recommendations [7], we will mainly refer to the most recent French recommendations for pediatric and adult-onset chronic non infectious uveitis [8].”

Also, I added:

Page 2, as a third paragraph in section 2 (Classical therapeutic approaches in JIA-associated chronic uveitis): “In children and adolescents with JIA and controlled uveitis who are tapering or discontinuing topical glucocorticoids, ophthalmologic monitoring within one month after each change of topical glucocorticoids is strongly recommended over monitoring less frequently [7].”

Page 2, the following paragraph now starts with: “In patients whose uveitis requires several months of local steroid therapy, the risk of side effects has led several panels of experts to recommend the introduction or change of systemic therapy rather than maintaining long-term local steroid treatment [6-8].”

Page 2, juste before Table 1: “In JIA patients with chronic anterior uveitis that is controlled on systemic therapy but remain on 1–2 drops/day of prednisolone acetate 1% (or equivalent), tapering topical glucocorticoids first is strongly recommended over systemic therapy.”

In Table 1, I deleted the link to the French rare disease network FAI2R

Page 4: “For several experts, in JIA patients with severe active chronic anterior uveitis and sight threating complications, starting methotrexate and a monoclonal anti-TNF antibody immediately is conditionally recommended over methotrexate as monotherapy [7-8].”

Page 4: “One of the ACR recommendations is that in JIA patients with active chronic anterior uveitis who have an inadequate response to one anti-TNF monoclonal antibody TNFi at standard JIA dose, escalating the dose and/or frequency to above-standard is conditionally recommended over switching to another anti-TNF monoclonal antibody TNFi [7]. However, this recommendation is only valid in the absence of anti-biodrug neutralizing antibodies”

Page 4, just after, as a first paragraph of “4.Other therapeutic options”: “In accordance with available evidence based medicine, one of the ACR recommendations is that in JIA patients with active chronic anterior uveitis who have failed a first anti-TNF monoclonal antibody at above-standard dose and/or frequency, changing to another anti-TNF monoclonal antibody is conditionally recommended over a biologic in another category [7].”

Page 5, after “Data for rituximab and Il-1 inhibitors are even more limited.” I added: “Nevertheless, one of the SHARE recommendations is that tocilizumab, rituximab and abatacept might be potential options for cases refractory to previous anti-TNF therapy [6].”

  1. In terms of steroid eyedrops, certainly dexamethasone may be used, but there are are topical steroids that should be mentioned as alternatives, especially since other countries, prednisone acetate 1% may be frequently used instead of dexamethasone.

I corrected accordingly in the revised version (I replaced “dexa” by “steroid” eye drops in the abstract and modified the sentence just before Table 1)

  1. Why stop immunosuppression after at least 1 year of quiescence? Although a low level of evidence/conditional recommendations, several other guidelines refer to at least 2 years. Authors should also reference the ongoing ADJUST trial (https://clinicaltrials.gov/ct2/show/NCT03816397).

Thank you for these important remarks. I took them into account by

a/ modyfing the sentence in the abstract

b/ indicating page 6: “Both the ACR and the European SHARE recommendations are in favor of at least 2 years of quiescence before tapering treatment intensity when uveitis is well controlled on DMARD and biologic systemic therapy only [6-7].”

c/ also now page 6: “The ongoing randomised controlled ADJUST trial (https://clinicaltrials.gov/ct2/show/NCT03816397) aims to assess the efficacy of discontinuing adalimumab after demonstrating control of JIA-associated uveitis for at least 12 months.”

  1. For the situation of “Marked macular edema on MTX and adalimumab therapy,” authors recommend replacing adalimumab by anti-IL-6 treatment, why not try infliximab?

As explained in this review and with some references, anti-IL-6 therapy might be particularly valuable in patients with marked macular edema.

Neverthess I agree it should be consolidated by more evidence based medicine.

  1. In general, do authors consider the use of other anti-TNF agents (eg infliximab) this should be discussed as the current portrayal of the of the article suggests that switching to anti IL-6 is the next step after adalimumab/MTX failure.

Thank you for pointing this; switching to another anti-TNF agent had indeed to be considered.

a/ I corrected: page 6: “In JIA patients with both arthritis and uveitis whose uveitis has been well controlled on adalimumab (and MTX if tolerated) but arthritis remains active, dosing adalimumab serum level and considering weekly injections is an option, moving to another biologic such as infliximab or an anti-IL-6 receptor antagonist that might be more effective on arthritis while also active on uveitis could be another option

b/ I added page 4 (first paragraph now of “Other therapeutic options): “In accordance with available evidence based medicine, one of the ACR recommendations is that in JIA patients with active chronic anterior uveitis who have failed a first anti-TNF monoclonal antibody at above-standard dose and/or frequency, changing to another anti-TNF monoclonal antibody is conditionally recommended over a biologic in another category [7]. According to the SHARE initiative, the use of anti-TNF treatment strategies (adalimumab>infliximab>golimumab) is recommended in patients with uveitis refractory/resistant to methotrexate and switching between different anti-TNF treatments might be valuable if uveitis is refractory to the first anti-TNF [6].”

Round 2

Reviewer 2 Report

Thank you for addressing my comments.